# USING GNNS TO MODEL BIASED CROWDSOURCED DATA FOR URBAN APPLICATIONS

## ABSTRACT

Graph neural networks (GNNs) are widely used to make predictions on graph-structured data in urban spatiotemporal forecasting applications, such as predicting infrastructure problems and weather events. In urban settings, nodes have a true latent state (e.g., street condition) that is sparsely observed (e.g., via government inspection ratings). We more frequently observe biased proxies for the latent state (e.g., via crowdsourced reports) that correlate with resident demographics. We introduce a GNN-based model that uses both unbiased rating data and biased reporting data to predict the true latent state. We show that our approach can both recover the latent state at each node and quantify the reporting biases. We apply our model to a case study of urban incidents using reporting data from New York City 311 complaints across 141 complaint types and rating data from government inspections. We show (i) that our model predicts more correlated ground truth latent states compared to prior work which trains models only on the biased reporting data, (ii) that our model's inferred reporting biases capture known demographic biases, and (iii) that our model's learned ratings capture correlations across locations and between complaint types. Especially in urban crowdsourcing applications, our analysis reveals a widely applicable approach for using GNNs and sparse ground truth data to estimate latent states.

## 1 INTRODUCTION

Graph neural networks (GNNs) have emerged as powerful and expressive models for making predictions on graph-structured data, especially for urban applications such as air quality monitoring, forecasting traffic flows, predicting housing prices, and modeling the spread of epidemics (Xie et al., 2019; Roy et al., 2021; Brimos et al., 2023; Yu et al., 2023; Zhan & Datta, 2024). In urban planning – our empirical setting – government officials often wish to know where urban incidents like rodents or floods truly occur so they can make downstream resource allocation decisions; however, this ground truth is typically unobserved and must be predicted. GNNs are a powerful tool to make these predictions, as they can naturally encode spatial correlations of the ground truth states across nodes in a graph (e.g., neighborhoods in a city). For example, if a flood has occurred in one neighborhood, the adjacent neighborhoods are also likely to be flooded.

Estimating latent ground truth for the hundreds of types of incidents that occur in a city is challenging. Nevertheless, there are two sources of information we can use, each with its own limitations. First, we observe the ground truth state via *government inspections* which generate *ratings* for neighborhoods. For example, New York City conducts street inspections for every street and rates them from 1-10. Importantly, these inspections are only conducted for some incident types and neighborhoods and are thus sparsely observed. These settings also often have another source of data: frequently observed, biased proxies of the latent state, e.g., via crowdsourced *reports* of incidents. Unlike ratings, reports are observed across all incident types, all neighborhoods, and multiple points in time. However, previous work has established that underreporting is pervasive and heterogeneous (Clark et al., 2020; Kontokosta & Hong, 2021; Agostini et al., 2024; Liu et al., 2024); in different neighborhoods that face similar incidents, residents often *report* those incidents at different rates. This presents an identifiability issue; if one neighborhood logs more reports than another, it is unclear whether the former has a worse ground truth or if given the same ground truth, the latter is less likely to report. Thus, reports may not accurately predict ground truth across all neighborhoods as the same ground truth state may have different reporting patterns across the city. For example, Casey et al. (2018)

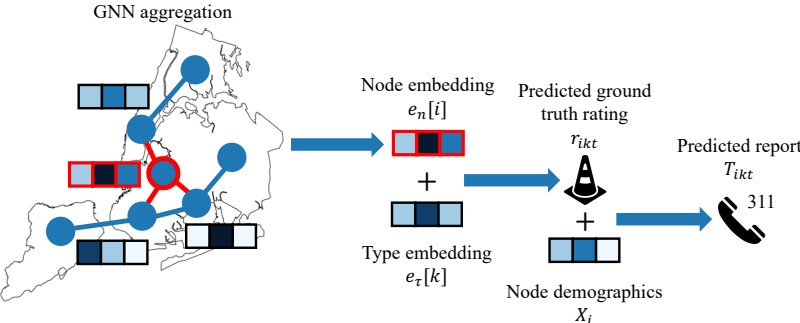

Figure 1: We use a GNN-based model to estimate two quantities: ground truth inspection ratings and reports of incidents. We model inspection ratings $r_{ikt}$ using node $i$'s learned node embedding $e_n[i]$ and type $k$'s learned type embedding $e_\tau[k]$. We model reports $T_{ikt}$ as a function of the rating $r_{ikt}$ and a set of node-specific demographic features $X_i$.

found that in Washington, D.C. crowdsourced reports on rodents did not accurately predict the outcome of inspections. Moreover, differences in reporting often correlate with demographics, so learning only from reporting data risks introducing bias against underserved populations.

We propose a novel GNN-based approach to capture the above characteristics of urban incident reporting – (i) high dimensionality: we have crowdsourced reports across many types (e.g., rodents, food poisoning, fallen trees, etc.) over time; (ii) frequently observed, biased reporting data: our reports capture whether incidents of each type were reported in each node in each granular time period; and (iii) sparsely observed, ground truth rating data: the city conducts periodic inspections which yield ratings for a sparse set of types, nodes, and time periods.

Our GNN-based approach jointly models both the the true latent state and the probability of a report for each node (neighborhood) across all incident types. Summarized in Figure 1, the model uses a GNN to capture spatial correlations in the ground truth state (e.g., street condition) and estimates how the reporting probability varies across nodes as a function of demographics. We train our model to simultaneously predict (i) ground truth ratings using learned node and type embeddings and (ii) how the likelihood of reporting varies by demographics, conditional on ground truth state. Since there is no way to distinguish between neighborhoods which truly do not have problems and neighborhoods that do not report them, estimating this model is impossible without some information to constrain or identify the ground truth state. We show on semi-synthetic data that by supplying the model with sparse ground truth rating data, we are able to identify which neighborhoods systematically underreport incidents. Thus, on types for which ground truth ratings are not observed, we are still able to infer the ground truth and correct for reporting biases.

We apply our model to a case study of New York City 311 complaints (crowdsourced reports), leveraging 55 million reports across 141 types over two years. We combine this with a carefully curated dataset of ground truth ratings which are sourced from 300k government inspections across 5 types in the same time frame. Using both semi-synthetic simulations and real data, we show that our approach can (i) estimate the ground truth inspection ratings and (ii) quantify the biases in the frequently observed reporting data, capturing the fact that different neighborhoods report similar issues at different rates. We find that the sparsely observed, ground-truth rating data and the frequently observed, biased reporting data both confer benefits. Using both the ground truth rating data and the reporting data, our model can infer ground truth ratings that are $2\times$ more correlated than a model that only uses reporting data. Additionally, our model predicts ratings that more plausibly reflect spatial correlations between nodes compared to a model that uses rating data alone. We also show that our model's inferred reporting biases align with known demographic patterns of underreporting and that our model's learned ratings capture both spatial correlations across neighborhoods and correlations across 311 complaint types. We will release code and data to replicate all experiments.

Although our primary application is to urban crowdsourcing, our approach is broadly applicable to other GNN prediction tasks where both sparsely observed, ground truth data and frequently observed, biased data are available. Specific application areas include other urban challenges (such as estimating air quality using both resident reports and sparse sensor measurements) and spatiotemporal processes (such as epidemic forecasting using both internet search data and sparse official health reports). Our

work also relates to prior work on semi-supervised learning on graphs with sparse ground truth labels (see related work in §2). Our analysis reveals a generalizable approach to using GNNs and sparse, ground truth data to identify latent states.

## 2 RELATED WORK

Our work relates to and extends several literatures: (1) urban spatiotemporal modeling and related methodology, including semi-supervised learning, (2) learning from noisy and human-reported data, and (3) GNN learning on noisy graphs. Our work is at the intersection of these literatures, augmenting biased labels with ground truth for graph learning applied to urban settings.

**Spatiotemporal modeling:** GNNs are a natural fit for high-dimensional spatiotemporal modeling in applications like traffic forecasting, epidemic forecasting, and molecular dynamics (Kapoor et al., 2020; Roy et al., 2021; Wang et al., 2022a;b; He et al., 2023; Pineda et al., 2023; Wu et al., 2024). Several works also design ways to encode spatiotemporal information in GNNs, including positional encoders (Klemmer et al., 2023), kriging convolutional networks (Appleby et al., 2020), and inductive kriging (Wu et al., 2021). Non-GNN-based spatiotemporal models, including Bayesian, clustering, and matrix factorization models, have also been used for urban issues like crime (Hu et al., 2018), pedestrian traffic (Zaouche & Bode, 2023), air pollution (Sarto et al., 2016), urban flow (Pan et al., 2019), and infrastructure monitoring (Budde et al., 2014).

For 311 complaints in particular, prior works have quantified underreporting of floods using spatiotemporal models (Agostini et al., 2024) and have more broadly quantified the geographic and demographic patterns of underreporting (Kontokosta et al., 2017; Wang et al., 2017; Kontokosta & Hong, 2021). Disparities in incident reporting rates lead to downstream inequities in resource allocation, so understanding the patterns of underreporting is crucial (Liu et al., 2024). Especially notable in relation to our work, one prior study in Washington, D.C. showed that 311 reports are poor predictors of ground truth ratings, in line with our hypotheses and findings (Casey et al., 2018). Our work extends this literature by proposing a specific approach to overcome the limitations of biased reporting data: leveraging sparse ground truth.

**Other methodological areas:** Our work ties in with three broad methodological areas: semi-supervised learning, learning from noisy and human-reported data, and GNN learning on noisy graphs. A long line of prior work has dealt with semi-supervised learning on graphs and noisy labels. Several prior works address our core issue of semi-supervised learning with sparsely observed, ground truth labels and frequently observed proxies. In some works, the proxy labels are the outputs of a machine learning model (Arazo et al., 2020) which are debiased to produce better predictions (Zhang et al., 2021). Wang et al. (2022c) tries to learn which proxies are reliable and upweights those which are predicted to be reliable. Our work extends this literature, by showing that models that use both reliable inspection ratings and less reliable crowdsourced reports improve upon models which only use the less reliable reports.

One common source of frequently observed, biased data is human behavior (such as crowdsourced reports). Prior work has shown that models which use this biased data can affect high-stakes decisions (Lum & Isaac, 2016; Obermeyer et al., 2019; Mullainathan & Obermeyer, 2021) and that crowdsourced labels vary across annotators and often correlate with demographics, indicating that different groups may perceive the same data (e.g., text, image, incident/problem) differently (Chakraborty et al., 2017; Zhang et al., 2017; Ding et al., 2022). There is also work that attempts to resolve a ground truth label from several annotations (Dawid & Skene, 1979; Bach et al., 2017).

Finally, several prior works provide methods for learning on noisy graphs. For instance, in social and citation networks the *graph itself* is often noisy and dynamic, leading to spurious correlations between nodes (Hamilton et al., 2017). Methods such as graph attention and causal regularization overcome these issues by filtering spurious correlations from causally relevant ones (Wang et al., 2019; 2022a; Wu et al., 2023). Other methods deal with noisy and sparse *labels* for nodes. Dai et al. (2021) and Qian et al. (2023) generate pseudolabels for nodes by aggregating information from the most similar labeled nodes. Crucially, these methods do not augment noisy node labels (i.e., reports) with ground truth data. Applications of the above techniques include inferring links in gene regulatory

networks (Singh et al., 2024), estimating the spread of infectious diseases (Tomy et al., 2022), and detecting vulnerabilities in software (Cheng et al., 2021).

Our work extends and combines insights from these areas: we use a GNN to model both sparsely observed, unbiased data and frequently observed, biased data generated from human behavior. We use both data sources to predict ground truth latent states and learn about reporting biases.

## 3  MODEL

**Approach overview:**   Our GNN model is summarized in Figure 1. The purpose of our model is to (i) estimate the true latent state of a particular incident type at a particular location – e.g., what is the true street condition in a particular neighborhood?; and (ii) to quantify biases in the observed reporting data – e.g., which neighborhoods systematically underreport incidents and how do reporting behaviors correlate with demographics? In many urban settings, models are fit using only the frequently observed reporting data, resulting in biased predictions (Xu et al., 2017; Casey et al., 2018; Li et al., 2020; Hacker et al., 2022). In contrast, our approach learns the true latent state using both the frequently observed, biased reporting data and the sparsely observed, unbiased rating data.

**Notation:**   Consider a network $G$ with $n$ nodes and adjacency matrix $E$. In our case study, nodes are indexed by $i$ and represent neighborhoods, and edges connect adjacent neighborhoods. Each node $i$ has features $X_i \in \mathbb{R}^D$, where $D$ is the number of features. These features include demographic factors that may influence reporting rates. There are $\tau$ incident types indexed by $k$ (e.g., rodents, floods, etc.). We index time by $t$ (e.g., weeks). We have two types of data: sparsely observed, unbiased true state measures (e.g., *inspection ratings*) and frequently observed, biased data (e.g., *crowdsourced reports*).

**Observed data:**   For some node/type/time tuples, we observe inspection ratings $r_{ikt} \in \mathbb{R}$. In our urban reporting case study, we source ratings from city government inspections for various government services (e.g., street ratings, park ratings, etc.). A lower inspection rating indicates a worse true state; e.g., a street with a lower rating has more damage. We normalize ($z$-score) the inspection ratings for each type across time and nodes. We use ratings from incident types for which inspections are conducted randomly and periodically (as opposed to in response to potentially biased reports) so that ratings are unbiased observations of the true latent state.[1] However, our observed inspection rating data is sparse and only available for a subset of nodes, types, and times.

We also observe reports of incidents $T_{ikt} \in \{0, 1\}$, where $T_{ikt} = 1$ indicates that an incident of type $k$ was reported for node $i$ at time $t$. In our urban reporting case study, we source reports from New York City's resident reporting system, NYC311. Reports are obtained from residents and are thus biased proxies of the true latent state that correlate with resident demographics.

Examples of $r_{ikt}$ and $T_{ikt}$ data exist in many settings. In environmental monitoring, $r_{ikt}$ may be geographically sparse sensor measurements of air quality, and $T_{ikt}$ may be resident reports of air quality. In epidemic forecasting, $r_{ikt}$ may be sparse health reports, and $T_{ikt}$ may be search data (Bauer & Aschenbruck, 2018; Chang et al., 2024).

**Model:**   We model ratings using a *node embedding* and a *type embedding*. Node $i$'s embedding $e_n[i] \in \mathbb{R}^{E_n}$, where $E_n$ is the embedding dimension, is a low-dimensional representation of a node and captures the node's attributes and position. The node embeddings are learned using a GNN (Kipf & Welling, 2017; Veličković et al., 2018), which is a deep learning model that leverages graph-structured data by iteratively aggregating and transforming features from neighboring nodes. Thus our node embeddings are *spatially correlated*, mirroring the correlation of true incident occurrence across neighborhoods. We also learn type $k$'s embedding $e_\tau[k] \in \mathbb{R}^{E_\tau}$, where $E_\tau$ is the embedding dimension. The type embedding is a low-dimensional representation of a type and captures the type's features, similarity to other types, and relationship to nodes in the graph. Thus our type embeddings capture correlations across types.

---

[1]In our empirical setting in New York City, many government agencies explicitly conduct proactive, regular inspections not in response to reports, in addition to also conducting inspections in response to reports (NYC Open Data, 2024e). We identify and filter out inspections made in response to reports. Details on how we filter inspections are provided in Appendix C.

More formally, we model the true latent state as follows:

$$\text{Predicted rating: } \hat{r}_{ikt} = e_n[i]^\top e_\tau[k]$$
$$\text{True inspection rating: } r_{ikt} \sim f_r(\cdot | \hat{r}_{ikt}) \tag{1}$$

The predicted rating $\hat{r}_{ikt}$ is estimated from node $i$'s embedding $e_n[i]$ and type $k$'s embedding $e_\tau[k]$. The true rating is drawn from a distribution $f_r$ parameterized by the predicted rating $\hat{r}_{ikt}$.

We model reports as follows:

$$\text{True report: } T_{ikt} \sim \text{Bernoulli}(\text{sigmoid}(\alpha_k r_{ikt} + \theta_k^\top X_i)) \tag{2}$$

Each report $T_{ikt}$ is drawn from a Bernoulli distribution parameterized by a logistic function of the true rating $r_{ikt}$ and node specific demographic features $X_i$, with unknown type-specific coefficients $\alpha_k \in \mathbb{R}$ and $\theta_k \in \mathbb{R}^D$. These coefficients are unique for each type which reflects that different incident types have different reporting characteristics, a claim we confirm on our real rating data.

We now discuss how we predict the probability of observing a report. For different node/type/time pairs $(i, k, t)$, we model the probability of observing a report differently depending on whether rating $r_{ikt}$ is observed and whether ratings for other nodes $i'$ for type $k$ are observed (i.e., whether ratings $r_{i'kt}$ are observed). Overall, there are three different cases that we consider:

*Case 1: Predicted probability of a report $\hat{P}(T_{ikt})$ when rating $r_{ikt}$ is observed.* In this case, we model the probability of observing a report as a function of the true, observed rating $r_{ikt}$, and we estimate type specific reporting coefficients $[\alpha_k, \theta_k]$ :

$$\text{Case 1: } \hat{P}(T_{ikt}) = \text{sigmoid}(\alpha_k r_{ikt} + \theta_k^\top X_i) \tag{3}$$

*Case 2: Predicted probabiity of a report $\hat{P}(T_{ikt})$ when rating $r_{ikt}$ is unobserved but ratings $r_{i'kt}$ for type $k$ are observed at other nodes $i'$.* In this case, we do not have access to node $i$'s true rating, so we model the probability of observing a report as a function of the *predicted rating $\hat{r}_{ikt}$* and type specific reporting coefficients $[\alpha_k, \theta_k]$. The type specific coefficients $[\alpha_k, \theta_k]$ are learned via equation 3 using the nodes $i'$ for which the ground truth ratings $r_{i'kt}$ are observed for type $k$.

$$\text{Case 2: } \hat{P}(T_{ikt}) = \text{sigmoid}(\alpha_k \hat{r}_{ikt} + \theta_k^\top X_i) \tag{4}$$

*Case 3: Predicted probabiity of a report $\hat{P}(T_{ikt})$ when rating $r_{ikt}$ is unobserved and no ratings for type $k$ are observed at any node.* We again do not have access to the true rating, so we model the probability of observing a report as a function of the *predicted rating $\hat{r}_{ikt}$*. We also cannot simultaneously learn the rating $r_{ikt}$ and the type specific reporting coefficients $[\alpha_k, \theta_k]$, thus we model the probability of observing a report as a function of the mean reporting coefficients across types with observed ratings $[\overline{\alpha}, \overline{\theta}]$.

$$\text{Case 3: } \hat{P}(T_{ikt}) = \text{sigmoid}(\overline{\alpha} \hat{r}_{ikt} + \overline{\theta}^\top X_i) \tag{5}$$

Learning a separate regression for each type $k$ allows us to recover type-specific reporting coefficients $[\alpha_k, \theta_k]$ which accounts for different types' reporting propensities. For instance, residents may be more likely to report rodents than a noise complaint. We implicitly assume here that the mean coefficients $[\overline{\alpha}, \overline{\theta}]$ are reasonable for types with unobserved ratings, i.e., the reporting coefficients transfer across types to some extent. We show in our semi-synthetic experiments that compared to a model trained on reporting data alone, our model, which uses both inspection rating data and reporting data, is able to predict ratings that are more correlated to the ground truth ratings, even for types for which the model does not observe any ground truth ratings.

We note that our approach easily extends to other parameterizations of $r_{ikt}$ and $T_{ikt}$. Thus, while our described model predicts constant ratings $\hat{r}_{ikt}$ and reporting probabilities $\hat{P}(T_{ikt})$ over time, our method generalizes to spatiotemporal GNN-based models. Full details on our model and learning procedure are provided in Appendix A.

**Loss function:**   To calculate our loss function we first separately evaluate our model's performance on predicting reports and ratings. Our final loss is a weighted sum of each of these individual loss components. More formally, the loss function consists of four parts:

| | Full model | Reports-only model | Ratings-only model |
|---|---|---|---|
| **Correlation on predicted reports** | 0.42 | 0.43 | – |
| **Correlation on predicted ratings** | 0.60 | 0.35 | 0.58 |

Table 1: **Semi-synthetic experimental results.** We compare our full model (which uses both reporting and rating data) to a reports-only and a ratings-only model. Compared to both baselines, our full model can estimate ratings without compromising accuracy in predicting reports. We calculate the correlation between our predicted probabilities of reports and the true probabilities for all node/type pairs. We calculate the correlation between our predicted ratings and the true ratings for all nodes and for all types with observed ratings. We report the median correlation across 5 synthetic datasets.

(i) *Report loss for data points with **unobserved** inspection ratings:* Binary cross entropy (BCE) between the true $T_{ikt}$ and predicted $\hat{P}(T_{ikt})$ for data points with unobserved inspection ratings.

$$\mathcal{L}_{\text{report unobserved}} = \sum_{ikt} \mathbb{1}\left(r_{ikt} \text{ is unobserved}\right) \cdot \text{BCE}(\hat{P}(T_{ikt}), T_{ikt}) \tag{6}$$

(ii) *Report loss for data points with **observed** inspection ratings:* BCE between the true $T_{ikt}$ and predicted $\hat{P}(T_{ikt})$ for data points with observed inspection ratings.

$$\mathcal{L}_{\text{report observed}} = \sum_{ikt} \mathbb{1}\left(r_{ikt} \text{ is observed}\right) \cdot \text{BCE}(\hat{P}(T_{ikt}), T_{ikt}) \tag{7}$$

(iii) *Rating loss:* Mean squared error (MSE) between the true rating $r_{ikt}$ and the predicted rating $\hat{r}_{ikt}$.

$$\mathcal{L}_{\text{rating}} = \sum_{ikt} \mathbb{1}\left(r_{ikt} \text{ is observed}\right) \cdot \text{MSE}(\hat{r}_{ikt}, r_{ikt}) \tag{8}$$

(iv) *Regularization loss:* $L^2$ norm of the predicted ratings $\hat{r}_{ikt}$. We include this loss to maintain stable training and prevent our predicted ratings from exploding.

$$\mathcal{L}_{\text{regularization}} = \sum_{ikt} L^2(\hat{r}_{ikt}) \tag{9}$$

The overall loss is as follows:

$$\mathcal{L} = \mathcal{L}_{\text{report unobserved}} + \gamma_1 \cdot \mathcal{L}_{\text{report observed}} + \gamma_2 \cdot \mathcal{L}_{\text{rating}} + \gamma_3 \cdot \mathcal{L}_{\text{regularization}} \tag{10}$$

We use weights $\gamma_1, \gamma_2, \gamma_3$ and fix the weight on $\mathcal{L}_{\text{report unobserved}}$ to 1. We select weights via a hyperparameter search maximizing the correlation of predicted reports and ratings. Details are in Appendix A.

## 4 SEMI-SYNTHETIC EXPERIMENTS

We now validate our proposed approach on semi-synthetic data. We verify that our model can accurately recover the true data-generating process (i.e., inspection ratings, crowdsourced reports, and the reporting coefficients) when our model is well-specified.

### 4.1 SEMI-SYNTHETIC DATA

For our semi-synthetic experiments, we use demographic features $X_i$ and reports $T_{ikt}$ from New York City 311 data (NYC Open Data, 2024a). We analyze all Census tracts[2] with valid demographic information ($n = 2292$ nodes), complaint types with a reporting frequency greater than 0.1% ($\tau = 141$ types), and all weeks in the two years from 2022 - 2023. $X_i$ represents 6 Census tract level demographic features and $T_{ikt} \in \{0, 1\}$ denotes whether at least one report of type $k$ was made in node $i$ during week $t$. In total we analyze more than 55 million reports. We then *generate* synthetic ratings $r_{ikt}$ so that we can compare our model's predictions against a known ground truth.

---

[2]A Census tract is a geographic region defined by the U.S. Census Bureau to analyze population data. On average, each census tract has thousands of inhabitants. There are 2326 total Census tracts in New York City.

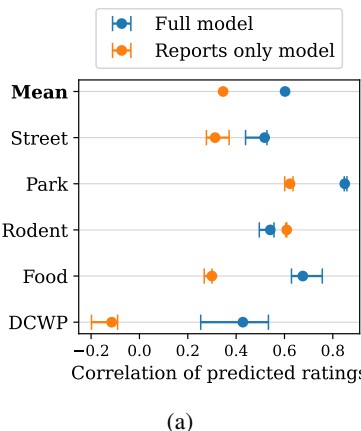
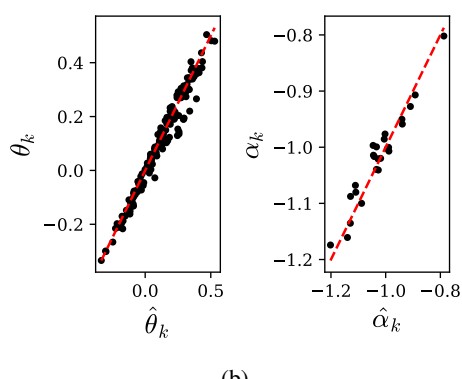

(a)  (b)

Figure 2: **Semi-synthetic experimental results.** Figure 2a: We show that our full model predicts more correlated ratings than a model that uses only reporting data. We calculate the correlation between the average predicted and true rating for each node/type pair. We show results for all types with observed inspection ratings and for the mean across these types. We plot the median correlation across 5 synthetic datasets. Error bars denote the range across the 5 trials. Figure 2b: We show that our model's predicted coefficients $[\hat{\theta}_k, \hat{\alpha}_k]$ match the true coefficients $[\theta_k, \alpha_k]$ for all types with observed inspection ratings. The red line indicates perfect prediction.

We generate synthetic inspection ratings $r_{ikt}$ by inverting equation 2:

$$r_{ikt} = \frac{1}{\alpha_k} \left( \text{logit}(\mathbb{E}_t(T_{ikt})) - \theta_k^\top X_i \right) \tag{11}$$

Here, $\mathbb{E}_t(T_{ikt})$ is defined as the empirical frequency of $T_{ikt}$ over all weeks in the dataset and $[\alpha_k, \theta_k]$ are type-specific reporting coefficients. Our synthetically generated inspection ratings $r_{ikt}$ aim to replicate our real inspection rating data described in §5.1. Thus, for each type $k$, we draw $\alpha_k$ and $\theta_k$ from a Gaussian with a standard deviation of $0.1$ and a mean equal to the average reporting coefficients predicted by a logistic regression model run on the real inspection rating data.[3] Full details on our synthetic inspection ratings are available in Appendix B.1.

We report results from 5 trials. For each trial, we draw a set of reporting coefficients $[\alpha_k, \theta_k]$; generate a new set of synthetic ratings; refit the model to that dataset; and evaluate the predicted ratings, reports, and reporting coefficients. We use a time-based split. We train on data from January 2022 to June 2023 and test on data from July 2023 to December 2023. We wish to assess the effect of using reports and ratings. Thus, we compare inferences from models with (i) both reports and ratings (*full model*), (ii) only reports (*reports-only model*), and (iii) only ratings (*ratings-only model*).

## 4.2 SEMI-SYNTHETIC RESULTS

Table 1 shows our results. Compared to the reports-only and ratings-only models, our full model estimates ratings without compromising performance in predicting reports. Across all types, the average correlation between our full model's predicted probability of a report $\hat{P}(T_{ikt})$ and the true probability $P(T_{ikt})$ is $0.42$. Across all types with observed ratings, the average correlation between our full model's predicted rating $\hat{r}_{ikt}$ and the true rating $r_{ikt}$ is $0.60$. We report RMSE results in Appendix Table 4.

Table 1 shows that compared to the reports-only model, the full model predicts ratings which better correlate with ground truth ($r = 0.60$ for the full model versus $0.35$ for the reports-only model), and the full model predicts reporting probabilities which are similarly correlated to the true probabilities ($r = 0.42$ for the full model versus $0.43$ for the reports-only model). Figure 2a breaks down the full model's improvement for each type with observed inspection ratings. In Appendix Figure 5 we also show that, compared to the reports-only model, our full model predicts ratings that are more correlated with ground truth even for types with unobserved ratings. This shows that ground truth

---

[3]We set the intercept of $\theta_k$ such that our generated $r_{ikt}$ are zero mean. Thus, our generated and real inspection ratings take on both negative and nonnegative values.

|  | Full model | Reports-only model | Ratings-only model |
|---|---|---|---|
| **Correlation on predicted reports** | 0.25 | 0.55 | – |
| **Correlation on predicted ratings** | 0.19 | 0.08 | 0.18 |

Table 2: **Real data results.** We compare our full model to a reports-only and a ratings-only model. Compared to both baselines, our full model can estimate ratings without overfitting to reports. We calculate the correlation between our predicted probabilities of reports and the true probabilities for all node/type pairs. We calculate the correlation between our predicted ratings and the true ratings for all nodes and for all types with observed ratings.

data for *observed* types are beneficial in predicting ratings for *unobserved* types. Importantly, this demonstrates that our model can learn ground truth characteristics that generalize *across* types, a key contribution over prior work which models only a single type at a time.

Next we compare our full model's predicted ratings to the ratings-only model's predicted ratings. When learning only from rating data, a model can only make predictions on the sparse set of types for which ratings are observed. Thus even though in Table 1, compared to the ratings-only model, the full model predicts ratings that are similarly correlated with the ground truth ratings ($r = 0.60$ for the full model versus $0.58$ for the ratings-only model), this comparison is only for types with *observed* ratings. For types with *unobserved* ratings, only our full model can generalize and predict ratings.

A final benefit of our model is that it recovers the true reporting coefficients $[\alpha_k, \theta_k]$, as shown in Figure 2b. In our semi-synthetic data, the reporting probability $P(T_{ikt})$ is defined as a logistic function of the node demographics $X_i$ and the true synthetic inspection rating $r_{ikt}$. One cannot identify both the reporting coefficients $[\alpha_k, \theta_k]$ *and* the inspection ratings $r_{ikt}$ using only crowdsourced reporting data. In particular, with only crowdsourced reporting data it is impossible to distinguish between a bad inspection rating that is never reported and a truly good inspection rating. Thus, to identify reporting coefficients, one must either use *both* rating and reporting data or make strong parametric assumptions (e.g., assume a shared reporting model across types).

Overall our semi-synthetic results show that our approach helps if our model is well-specified. In the next section, we assess on real inspection rating data.

## 5 REAL-WORLD CASE STUDY: NEW YORK CITY RESIDENT REPORTING

In the following sections, we describe our experimental set up (§5.1), validate our fitted model (§5.2), and investigate clusterings of our learned ratings (§5.3).

### 5.1 EXPERIMENTAL SETUP

As in the semi-synthetic experiments, we use 55 million NYC 311 reports across 2292 nodes, 141 types, and two years. We collect ratings from government inspection data for five complaint types: (i) street conditions (NYC Open Data, 2023), (ii) park maintenance or facility conditions (NYC Open Data, 2024c), (iii) rodents (NYC Open Data, 2024e), (iv) food establishment/mobile food vendor/food poisoning (NYC Open Data, 2024d), and (v) DCWP consumer complaints (NYC Open Data, 2024b). We process the inspection data to remove any inspections triggered by 311 reports. Details are in Appendix C.

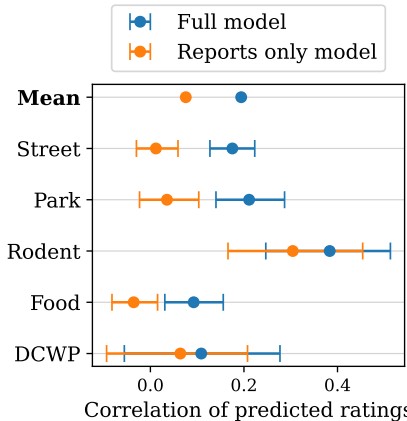

Figure 3: On real data, our full model predicts more correlated ratings than the reports-only model. Results are shown for all types with observed ratings and the mean across these types. We plot bootstrapped mean correlation and 95% CIs over node/type pairs.

We split our data into a train and test set using a time-based split, as is standard in urban planning (Yu et al., 2018; Farahmand et al., 2023; Huang et al., 2023; Agostini et al., 2024). We train our model on data from January 2022 to June 2023 and we test on data from July 2023 to December 2023.

## 5.2 VALIDATING THE MODEL

Prediction in real data is more challenging than in our semi-synthetic setting due to model misspecification. We model the probability of a report as a logistic function of demographics and true ratings, which allows us to quantify how reporting rates vary by demographics. But in reality, it is likely that reports are generated by a more complex function with more complex inputs. Nevertheless, our model's predicted ratings and reports still correlate with ground truth. As shown in Table 2, across all types, the average correlation between our full model's predicted probability of a report $\hat{P}(T_{ikt})$ and the true probability of a report $P(T_{ikt})$ is 0.25. Across all types with observed inspection ratings, the average correlation between our full model's predicted rating $\hat{r}_{ikt}$ and the true rating $r_{ikt}$ is 0.19. We report RMSE results in Appendix Table 5.

**Compared to the reports-only model, our full model's predicted ratings are more correlated with the ground truth ratings:** Table 2 shows that compared to the reports-only model, the full model's predicted ratings are more correlated with ground truth ratings ($r = 0.19$ for the full model versus 0.08 for the reports-only model). We also see that, compared to the reports-only model, the full model's predicted probabilities of reports are less correlated with ground truth probabilities ($r = 0.25$ for the full model versus 0.55 for the reports-only model). This is because the reports-only model overfits to the biased reporting data, evidenced by the model's large disparity in performance between predicted reports and ratings. Overall, our priority is to accurately predict ratings, and we find in both semi-synthetic and real data, that compared to a model that uses reporting data alone, our full model's predicted ratings are more correlated with ground truth ratings. Importantly, this highlights a key contribution of our model which leverages sparse, unbiased rating data over prior work which only learns from biased reporting data. Figure 3 breaks down the full model's improvement in predicting ratings for each type with observed inspection ratings.

Table 2 shows that compared to the ratings-only model, the full model's predicted ratings achieve comparable correlations with ground truth ($r = 0.19$ for the full model versus 0.18 for the ratings-only model). Investigating whether refinements of our model can yield improved predictions by incorporating reporting data represents a promising direction for future work.

$\theta_k$ **captures known demographic predictors of underreporting:** $\theta_k$ measures the contribution of each demographic feature in $X_i$ to the reporting rate. We estimate $\theta_k$ by fitting univariate variants of our full model. Each univariate model is trained on both rating and reporting data, but only uses one demographic feature. We run a separate univariate model for each demographic feature in $X_i$. Table 3 shows that

| Covariate | Mean coefficient |
|---|---|
| log(Population density) | 0.27 |
| Bachelors degree population | 0.16 |
| Households occupied by renter | 0.13 |
| log(Median income) | 0.12 |
| White population | 0.08 |
| Median age | 0.06 |
| True inspection rating | −0.20 |

Table 3: **Univariate demographic coefficients.** We report the average predicted univariate demographic coefficients across types with observed ratings. The estimated coefficients capture known demographic factors: tracts that are more dense, more educated, have a higher income, have more white residents, or are older are more likely to report incidents. We also report the average coefficient on the true inspection rating across all univariate models. Tracts that have lower ratings are more likely to be reported.

the inferred coefficients capture known demographic predictors of underreporting. Consistent with prior work, tracts that are more dense, more educated, have a higher income, have more white residents, or are older are more likely to report incidents (Kontokosta & Hong, 2021; Agostini et al., 2024; Liu et al., 2024). We report the coefficients predicted by a multivariate model in Table 6.

## 5.3 CLUSTERING NODES AND INCIDENT TYPES

**Predicted ratings are spatially correlated:** For each node $i$, we create a vector $\mathbf{r}_i = \{r_{ikt}\}_{k=1}^{\tau}$ of ratings over all types $k$. We use each node's $\mathbf{r}_i$ vector to cluster the nodes into 4 groups. We find

that the predicted clusters are spatially correlated and demographically distinct. Figure 4 shows that our clusters are spatially correlated, e.g., there is a clear spatial separation. The clusters correlate with New York City (NYC) borough lines, e.g., Manhattan falls mostly into cluster 0 and the Bronx falls mostly into cluster 3. Each NYC borough functions as a separate administrative area and corresponds to significant socioeconomic and other demographic differences. We similarly find significant demographic differences between the nodes in each of our predicted clusters, and we report the statistically significant differences in Appendix Table 7.

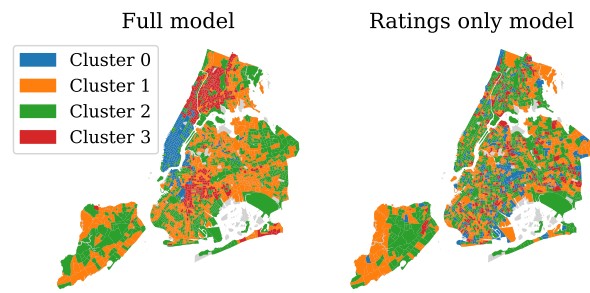

We compare our full model's clustering to the ratings-only model's clustering. Figure 4 shows that our full model learns more spatially correlated ratings than the ratings-only model. Many urban phenomena are spatially correlated, e.g. if a flood occurs in one neighborhood, it is likely that adjacent neighborhoods have also flooded. Prior work has used the spatial correlation of ground truth data as an identification approach (Agostini et al., 2024). Thus, while adding reporting data does not improve our rating predictions, it allows the full model to predict more reasonable ratings compared to the ratings-only model.

Figure 4: Using each node's vector of learned ratings over types, we cluster nodes into 5 groups using a $k$ means clustering algorithm. Our model which learns from both reports and ratings predict more spatially clustered ratings than a model which learns only from ratings.

**Ratings capture correlations between complaint types:** For each type $k$, we create a vector $\mathbf{r}_k = \{r_{ikt}\}_{i=1}^n$ of ratings over all nodes $i$ to cluster the types into 8 groups. We find that each group contains a coherent cluster of types, and in Appendix Table 8 we describe and list the types captured by each cluster. Additionally, Appendix Figure 6 shows that the dimension of highest variability (i.e., first PCA dimension) of the $\mathbf{r}_k$ vectors captures type frequency (i.e., $\mathbb{E}_{it}[T_{ikt}]$).

## 6 DISCUSSION

We address the challenging problem of estimating graph neural networks (GNNs) in settings where we observe biased outcome data. In these settings, nodes have a true latent state that is sparsely observed (e.g., via inspection ratings). We often also frequently observe biased proxies of the latent states (e.g., via crowdsourced reports). We propose a GNN-based model that uses frequently observed, biased reporting data and sparsely observed, unbiased rating data. We apply our model to New York City 311 data and show that (i) our model makes better predictions of the ground truth latent state compared to a baseline model trained only on reporting data, (ii) our model's inferred reporting biases capture known demographic factors of underreporting, and (iii) our model's learned ratings capture correlation between nodes and 311 complaint types.

Our model opens several avenues for additional applications and future work. Here we experimented with a particular model for reporting propensity; one line of future work could investigate whether other methods of incorporating reporting data can produce more accurate rating predictions. Another natural direction is to apply our model to other urban settings with biased reporting data. As discussed in this paper, reporting biases often correlate with demographics, so learning only from reporting data risks introducing bias against underserved populations. Beyond urban applications, other GNN tasks may also have sparsely observed ground truth and frequently observed biased proxies. In such settings, our model is a key advance in using GNNs to estimate ground truth latent states.

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
