## A    FURTHER DETAILS ON MODEL

We learn node embeddings using a graph neural network (GNN) (Kipf & Welling, 2017; Veličković et al., 2018), which is a deep learning model that leverages graph-structured data by iteratively aggregating and transforming feature information from neighboring nodes. The GNN takes in as inputs the graph $G$ and a set of features for each node $i$. We use one-hot node features which is common in settings like ours without natural node features (Cui et al., 2022). We learn the type embeddings using a linear layer with one-hot type feature inputs.

The details of our model architecture are as follows: We use a 2 layer GNN where each layer consists of a graph convolution, leaky ReLU activation, and batch normalization. We use an intermediate dimension equal to the number of nodes $n = 2292$ and an embedding dimension of $E_n = E_\tau = 50$.

We batch our data such that data points with observed ratings and unobserved ratings are batched separately. During training, we freeze the reporting model for batches for which there are no observed ratings (i.e., we learn the reporting coefficients only from types for which ratings are observed).

We conduct a hyperparameter search over the loss weights $\gamma_1, \gamma_2, \gamma_3$, embedding dimension sizes, number of layers, batch size, and learning rate using Weights and Biases on a validation set. We select the set of hyperparameters that maximize the correlation of predicted reports and ratings. Based on the hyperparameter search, we run experiments with a learning rate of $0.01$ and a batch size of $16000$. Our full model uses weights $\gamma_1 = 20, \gamma_2 = 1, \gamma_3 = 10^{-6}$. All experiments are conducted on a cluster with access to NVIDIA A100 and A6000 GPUs. Our model can comfortably train on one GPU.

In our experiments, we wish to assess the effect of using reports and ratings. Thus, we compare inferences from models with (i) both reports and ratings (*full model*), (ii) only reports (*reports-only model*), and (iii) only ratings (*ratings-only model*). The full model uses both reporting and rating data and all demographic coefficients. Its hyperparameters are set to the specifications listed above. The reports-only and ratings-only models are identical to the full model, except for their loss. The reports-only model sets a weight of 0 on the loss terms that evaluate against ground truth reports $\mathcal{L}_{\text{report unobserved}}, \mathcal{L}_{\text{report observed}}$. The ratings-only model sets a weight of 0 on the loss term that evaluates against ground truth ratings $\mathcal{L}_{\text{rating}}$.

## B    FURTHER DETAILS ON SEMI-SYNTHETIC EXPERIMENTS

### B.1    SEMI-SYNTHETIC DATA

We generate synthetic inspection ratings $r_{ikt}$ using equation 11. We separately generate ratings for the train and test split. For example, for the train split, $\mathbb{E}_t(T_{ikt})$ is defined as the empirical frequency of $T_{ikt}$ over all weeks in the train time period. We draw $\alpha_k$ and $\theta_k$ from a Gaussian. The mean of the Gaussian is calculated as follows: We take our real rating data, and separately for each type fit a logisitic regression predicting reports from demographics and the ground truth rating ($T_{ikt} \sim X_i, r_{ikt}$). We set the mean $\alpha_k$ and $\theta_k$ to be the mean coefficients predicted across these type-specific logisitc regressions. We set the intercept such that the ratings are zero mean. Thus, our generated and real inspection ratings take on both negative and nonnegative values.

### B.2    SEMI-SYNTHETIC RESULTS

|  | Full model | Reports-only model | Ratings-only model |
|---|---|---|---|
| **RMSE on predicted reports** | 0.08 | 0.06 | – |
| **RMSE on predicted ratings** | 1.01 | – | 1.01 |

Table 4: **Semisynthetic data RMSE results.** We compare our full model to a reports-only and a ratings-only model. Compared to both baselines, our full model can estimate ratings without compromising accuracy in predicting reports. We calculate the RMSE between our predicted probabilities of reports and the true probabilities for all node/type pairs. We calculate the RMSE between our predicted ratings and the true ratings for all nodes and for all types with observed ratings. We report the median correlation across 5 synthetic datasets.

We evaluate our predicted reports and ratings using both *correlation* and *root mean squared error (RMSE)*.

**Correlation results:** We evaluate reports by calculating the correlation between our models predicted probability of a report and the average true report across each node/type pair. In other words, we calculate $\text{corr}(\hat{P}(T_{ikt}), \mathbf{E}_t[T_{ikt}])$. We evaluate ratings by calculating the correlation between our models predicted rating and the average true rating across each node/type pair. In other words, we calculate $\text{corr}(\hat{r}_{ikt}, \mathbf{E}_t[r_{ikt}])$.

In Table 1, we calculate the average correlation on reports across all node/type pairs and the average correlation on ratings across node/type pairs with observed ratings. The ratings-only model only predicts ratings, so we cannot evaluate its performance on predicting reports. Similarly, the reports-only model only predicts probabilities of reports. Thus in order to estimate the reports-only model's correlation on ratings we use the predicted probability of a report as a proxy for rating and evaluate $\text{corr}(\hat{P}(T_{ikt}), \mathbf{E}_t[r_{ikt}])$.

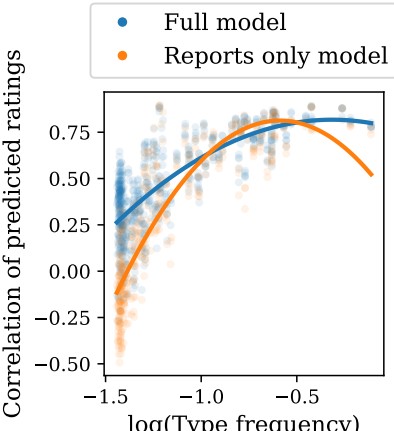

Figure 5: We evaluate our model's performance in predicting ratings across type frequencies. We measure type frequency as $\mathbb{E}_{it}[T_{ikt}]$. Particularly for rare types, compared to the reports-only model, our full model, which uses both reporting and rating data, predicts more correlated ratings. We show results for all types that the model *does not* observe ratings for. We plot the median across 5 synthetic datasets.

In Figure 2a, we calculate the correlation on reports for each type with observed ratings separately. In Figure 5, we calculate the correlation on reports for each type with unobserved ratings. In both cases, compared to the reports-only model, we find that our full model predicts ratings that are more correlated with the ground truth.

**RMSE results:** We evaluate reports by calculating the RMSE between our models predicted probability of a report and the average true report across each node/type pair. In other words, we calculate $\text{RMSE}(\hat{P}(T_{ikt}), \mathbf{E}_t[T_{ikt}])$. We evaluate ratings by calculating the RMSE between our models predicted rating and the average true rating across each node/type pair. In other words, we calculate $\text{RMSE}(\hat{r}_{ikt}, \mathbf{E}_t[r_{ikt}])$.

In Table 4, we calculate the average RMSE on reports across all node/type pairs and the average RMSE on ratings across node/type pairs with observed ratings. We report the median RMSE across 5 synthetic datasets. We note that the ratings-only model only predicts ratings. Therefore, we cannot evaluate its performance on predicting reports. Similarly, the reports-only model only predicts probabilities of reports. Therefore, we cannot evaluate its performance on predicting ratings. Note that unlike for correlation, when calculating RMSE, we *cannot* use a proxy for rating (e.g., $\hat{P}(T_{ikt})$).

|  | Full model | Reports-only model | Ratings-only model |
|---|---|---|---|
| **RMSE on predicted reports** | 0.11 | 0.06 | – |
| **RMSE on predicted ratings** | 0.83 | – | 0.84 |

Table 5: **Real data RMSE results.** We compare our full model to a reports-only and a ratings-only model. Compared to both baselines, our full model can estimate ratings without compromising accuracy in predicting reports. We calculate the RMSE between our predicted probabilities of reports and the true probabilities for all node/type pairs. We calculate the RMSE between our predicted ratings and the true ratings for all nodes and for all types with observed ratings.

## C    FURTHER DETAILS ON REAL DATA

**Details on processing real reporting data:**    We use reports $T_{ikt}$ from New York City 311 data (NYC Open Data, 2024a). We analyze all Census tracts with valid demographic information ($n = 2292$ nodes), complaint types with a reporting frequency greater than 0.1% ($\tau = 141$ types), and all weeks in the two years from 2022 - 2023. $T_{ikt} \in \{0, 1\}$ denotes whether at least one report of type $k$ was made in node $i$ during week $t$. In total we analyze more than 55 million reports.

**Feature processing:**    We include demographic features collected for each Census tract. The full list of features that we include is: log population density, percentage of population with a bachelors degree, percentage of households occupied by a renter, log median income, percentage of population that is white, and median age. We normalize all features to have mean 0 and standard deviation 1.

**Details on processing real rating data:** We collect ratings from government inspection data for five complaint types: (i) street conditions (NYC Open Data, 2023), (ii) park maintenance or facility conditions (NYC Open Data, 2024c), (iii) rodents (NYC Open Data, 2024e), (iv) food establishment/mobile food vendor/food poisoning (NYC Open Data, 2024d), and (v) DCWP consumer complaints (NYC Open Data, 2024b). Each rating is for a fine-grained unit within a Census tract. Street ratings are for street segments; park ratings are for parks; rodent ratings are averaged over each Borough-Block-Lot (BBL); food ratings are averaged over each BBL; and DCWP ratings are averaged over each Census block. We match each fine-grained rat-

| Covariate | Mean coefficient |
|---|---|
| Bachelors degree population | 0.28 |
| Households occupied by renter | 0.24 |
| log(Population density) | 0.20 |
| Median age | 0.13 |
| White population | −0.08 |
| log(Median income) | −0.10 |
| True inspection rating | −0.20 |

Table 6: **Multivariate reporting coefficients.** We report the average predicted multivariate demographic coefficients across types with observed ratings. The estimated coefficients capture known demographic factors: tracts that are more dense, more educated, or are older are more likely to report incidents. We also report the coefficient on the true inspection rating. Tracts that have lower ratings are more likely to be reported.

ing to its corresponding fine-grained report (i.e., reports in that same street segment). For rodents, food, and DCWP the matching is done directly (i.e., we match the aggregated rodent rating for a BBL to the aggregated report for the same BBL). For streets and parks, we run a distance heuristic to complete the matching. We match each rating with its nearest report. If the nearest report is above a certain distance threshold, we filter out the rating (consider it unobserved). Within the same tract, all fine-grained ratings and reports are provided to the model and are mapped to the same node's embedding, as well as the corresponding type's embedding.

We also process the inspection data to remove any inspections triggered by 311 reports. The rodent inspection data dictionary states that DOHMH inspectors run both random inspections and inspections triggered by 311 reports NYC Open Data (2024e). It is also stated that the random inspections occur block by block. The inspection data is not labeled as random versus 311 initiated, thus we run a heuristic to identify inspections triggered by 311 reports. We calculate the number of inspections that occur each week in each Census tract. We filter out all inspections that fall in tracts under the 50th percentile. Inspection data for the other types are described to be purely random.

## D    FURTHER DETAILS ON THE REAL-WORLD CASE STUDY

**Real data results**    We report our model's correlation on predicted reports and ratings in Table 2. In Table 5, we report our model's RMSE on predicted reports and ratings.

**Predicted demographic coefficients:**    In Table 3 we report the demographic coefficients predicted by univariate models. In Table 6 we report the demographic coefficients predicted by a multivariate model.

**Clustered nodes are demographically distinct.**    For each node $i$, we create a vector $\mathbf{r}_i = \{r_{ikt}\}_{k=1}^{\tau}$ of ratings over all types $k$. We use each node's $\mathbf{r}_i$ vector to cluster the nodes into 4 groups. We

| Cluster | 0 | 1 | 2 | 3 |
|---|---|---|---|---|
| Race:Non-Hispanic White | **55%** | 29% | 34% | 12% |
| Race:Asian | 16% | **18%** | **18%** | 5% |
| Race:African-American | 8% | 19% | 22% | **35%** |
| Households occupied by renter | 72% | 60% | 46% | **87%** |
| Bachelors degree | **71%** | 33% | 36% | 26% |
| Population | **5,500** | 3,900 | 2,600 | 5,200 |
| Median income | **120,000** | 71,000 | 73,000 | 47,000 |
| Median age | 37 | 38 | **40** | 35 |

Table 7: **Clustering ratings for each node**. We find that the clustering correlates with differences in demographics. All differences between clusters are statistically significant ($p < 0.001$, ANOVA test). The largest value in each row is shown in bold.

find that the predicted clusters are spatially correlated and demographically distinct. We report the statistically significant differences in demogarphics for each cluster in Table 7.

**Clustering ratings for each type:** For each type $k$, we create a vector $\mathbf{r}_k = \{r_{ikt}\}_{i=1}^n$ of ratings over all nodes $i$ to cluster the types into 8 groups. We find that each group contains a coherent cluster of types, and in Table 8 we describe and list the types captured by each cluster. Additionally, Figure 6 shows that the dimension of highest variability (i.e., first PCA dimension) of the $\mathbf{r}_k$ vectors captures type frequency (i.e., $\mathbb{E}_{it}[T_{ikt}]$).

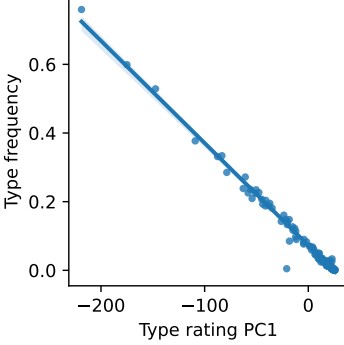

Figure 6: Each type's learned ratings over nodes capture type frequency information. In particular, the dimension of highest variance of our type ratings (PC1) has a high correlation with type reporting frequency $\mathbb{E}_k[T_{ikt}]$.

Table 8: **Ratings capture correlations between 311 complaint types:** Using each type's vector of learned ratings over nodes, we cluster types into 8 groups using a $k$ means clustering algorithm. We manually assign a succinct cluster description to each group. We find that the clusters group similar types together.

| Cluster Description | Complaint Types |
|---|---|
| Noise and Public Assistance Issues | Consumer Complaint (DCA) 
 Noise (DEP) 
 Homeless Person Assistance (DHS) 
 Traffic Signal Condition (DOT) 
 Encampment (NYPD) 
 Noise – Commercial (NYPD) 
 Noise – Vehicle (NYPD) 
 For Hire Vehicle Complaint (TLC) |
| Residential and Parking Violations | Dirty Condition (DSNY) 
 Missed Collection (DSNY) 
 Heat/Hot Water (HPD) 
 Unsanitary Condition (HPD) 
 Blocked Driveway (NYPD) 
 Illegal Parking (NYPD) 
 Noise – Residential (NYPD) 
 Noise – Street/Sidewalk (NYPD) |
| Housing Maintenance | Appliance (HPD) 
 Door/Window (HPD) 
 Electric (HPD) 
 Flooring/Stairs (HPD) 
 General (HPD) 
 Paint/Plaster (HPD) 
 Plumbing (HPD) 
 Water Leak (HPD) |
| Street and Vehicle Conditions | Sewer (DEP) 
 Water System (DEP) 
 General Construction/Plumbing (DOB) 
 Rodent (DOHMH) 
 Sidewalk Condition (DOT) 
 Street Light Condition (DOT) 
 Damaged Tree (DPR) 
 Derelict Vehicles (DSNY) 
 Illegal Dumping (DSNY) 
 Abandoned Vehicle (NYPD) |
| Environmental and Building Operation Concerns | Air Quality (DEP) 
 Lead (DEP) 
 Building/Use (DOB) 
 Elevator (DOB) 
 Curb Condition (DOT) 
 Street Sign – Damaged (DOT) 
 Dead/Dying Tree (DPR) 
 New Tree Request (DPR) 
 Overgrown Tree/Branches (DPR) 
 Root/Sewer/Sidewalk Condition (DPR) 
 Dead Animal (DSNY) 
 Electronics Waste Appointment (DSNY) 
 Obstruction (DSNY) 
 Residential Disposal Complaint (DSNY) 
 Street Sweeping Complaint (DSNY) 
 Safety (HPD) 
 Non-Emergency Police Matter (NYPD) |

| Health and Safety | Asbestos (DEP) |
| | AHV Inspection Unit (DOB) |
| | BEST/Site Safety (DOB) |
| | Scaffold Safety (DOB) |
| | Asbestos (DOHMH) |
| | Beach/Pool/Sauna Complaint (DOHMH) |
| | Construction Lead Dust (DOHMH) |
| | Drinking Water (DOHMH) |
| | Illegal Animal Kept as Pet (DOHMH) |
| | Indoor Sewage (DOHMH) |
| | Mold (DOHMH) |
| | Mosquitoes (DOHMH) |
| | Pet Shop (DOHMH) |
| | Poison Ivy (DOHMH) |
| | Tattooing (DOHMH) |
| | Bike Rack Condition (DOT) |
| | Bus Stop Shelter Placement (DOT) |
| | DEP Street Condition (DOT) |
| | E-Scooter (DOT) |
| | Uprooted Stump (DPR) |
| | Wood Pile Remaining (DPR) |
| | Adopt-A-Basket (DSNY) |
| | Seasonal Collection (DSNY) |
| | Outside Building (HPD) |
| | Sewer (NYC311-PRD) |
| | Water System (NYC311-PRD) |
| | Disorderly Youth (NYPD) |
| | For Hire Vehicle Report (TLC) |
| | Green Taxi Complaint (TLC) |
| | Taxi Report (TLC) |
| Public Space and Community Violations | Consumer Complaint (DCWP) |
| | Encampment (DHS) |
| | Boilers (DOB) |
| | Emergency Response Team (ERT) (DOB) |
| | Real Time Enforcement (DOB) |
| | Special Projects Inspection Team (SPIT) (DOB) |
| | Indoor Air Quality (DOHMH) |
| | Smoking (DOHMH) |
| | Broken Parking Meter (DOT) |
| | Outdoor Dining (DOT) |
| | Street Sign - Dangling (DOT) |
| | Street Sign - Missing (DOT) |
| | Animal in a Park (DPR) |
| | Illegal Tree Damage (DPR) |
| | Maintenance or Facility (DPR) |
| | Violation of Park Rules (DPR) |
| | Commercial Disposal Complaint (DSNY) |
| | Graffiti (DSNY) |
| | Litter Basket Request (DSNY) |
| | Noise - Helicopter (EDC) |
| | Animal-Abuse (NYPD) |
| | Bike/Roller/Skate Chronic (NYPD) |
| | Drug Activity (NYPD) |
| | Graffiti (NYPD) |
| | Illegal Fireworks (NYPD) |
| | Noise - Park (NYPD) |
| | Panhandling (NYPD) |
| | Traffic (NYPD) |
| | Lost Property (TLC) |
| | Taxi Complaint (TLC) |

| | |
|---|---|
| Sanitation and Water | Hazardous Materials (DEP) |
| | Industrial Waste (DEP) |
| | Water Conservation (DEP) |
| | Water Quality (DEP) |
| | Electrical (DOB) |
| | Investigations and Discipline (IAD) (DOB) |
| | Plumbing (DOB) |
| | School Maintenance (DOE) |
| | Day Care (DOHMH) |
| | Face Covering Violation (DOHMH) |
| | Food Establishment (DOHMH) |
| | Harboring Bees/Wasps (DOHMH) |
| | Non-Residential Heat (DOHMH) |
| | Standing Water (DOHMH) |
| | Unleashed Dog (DOHMH) |
| | Unsanitary Animal Pvt Property (DOHMH) |
| | Unsanitary Pigeon Condition (DOHMH) |
| | Bus Stop Shelter Complaint (DOT) |
| | Street Condition (DOT) |
| | Abandoned Bike (DSNY) |
| | Dumpster Complaint (DSNY) |
| | Illegal Posting (DSNY) |
| | Litter Basket Complaint (DSNY) |
| | Lot Condition (DSNY) |
| | Sanitation Worker or Vehicle Complaint (DSNY) |
| | Snow or Ice (DSNY) |
| | Elevator (HPD) |
| | Drinking (NYPD) |
| | Noise – House of Worship (NYPD) |
| | Urinating in Public (NYPD) |