# OpenReview forum: "Using GNNs to Model Biased Crowdsourced Data for Urban Applications"
_ICLR.cc/2025/Conference — ICLR 2025 Conference Withdrawn Submission_

### Official Review · Reviewer_2jyy · 2024-11-01

**Soundness:** 1
**Presentation:** 2
**Contribution:** 1
**Rating:** 3
**Confidence:** 3

**Summary:**

This paper proposes employing GNNs to model biased crowdsourced data for urban applications. Specifically, it leverages both unbiased rating data and biased reporting data to predict the true latent state in urban settings through a GNN framework. The authors conducted experiments using both a semi-synthetic dataset and a case study dataset to demonstrate the effectiveness of the proposed framework, showing that the model can recover the latent state and quantify reporting biases.

**Strengths:**

1. The research question is interesting.
2. Conducted experiments on both semi-synthetic and real-world datasets.

**Weaknesses:**

1. The proposed method is based on a standard GNN architecture without introducing any technical innovations.
2. In Eq. (1), $r_{ikt}$ is sampled from a distribution based on the predicted rating; however, the authors refer to it as the true inspection rating, which is confusing. In addition, how is the predicted rating generated—by multiplying $e_n$ and $e_t$?
3. The authors argue that their method can be applied to air quality and epidemic forecasting; however, they do not provide further discussion or experiments to support this claim.
4. The reports-only and ratings-only models are variants of the proposed model, so using them as baselines may not be appropriate. As you mention in your related works, there are several existing studies tackling urban computing tasks. Why not use them as baselines? For example,
[1] Hamed Farahmand, Yuanchang Xu, and Ali Mostafavi. A spatial–temporal graph deep learning model for urban flood nowcasting leveraging heterogeneous community features. Scientific Reports, 13 (1):6768, 2023
[2] Tao Hu, Xinyan Zhu, Lian Duan, and Wei Guo. Urban crime prediction based on spatio-temporal bayesian model. PloS one, 13(10):e0206215, 2018.

**Questions:**

Ref. weaknesses.

---

### Official Review · Reviewer_KUgX · 2024-11-03

**Soundness:** 3
**Presentation:** 2
**Contribution:** 3
**Rating:** 6
**Confidence:** 3

**Summary:**

The paper proposes a graph neural network model that addresses the challenge of using biased crowdsourced data to predict true urban incidents like infrastructure issues. In urban environments, ground truth states are sparsely observed through government inspections, while crowdsourced reports, which are more frequent, tend to be biased and influenced by demographics. The authors introduce a GNN framework that leverages both these sources: sparse, unbiased inspection data and frequent, biased reports. Their model jointly predicts the true latent states and quantifies reporting biases, achieving better accuracy in urban predictions compared to existing methods that only use reporting data. The paper presents a case study using New York City's 311 data, showing improved predictions and a more nuanced understanding of demographic biases in reporting.

**Strengths:**

- This paper presents a novel approach by leveraging graph neural networks to integrate sparse, unbiased inspection data with frequent but biased crowdsourced reports. The ability to quantify and adjust for reporting biases based on demographic features is new and addresses a significant gap in current urban modeling techniques. The proposed method shows potential to overcome limitations that arise when relying solely on biased data, making a meaningful contribution to the field.
- The model's ability to quantify reporting biases and use them to improve predictions adds significant value. The authors provide clear evidence that their model outperforms traditional approaches in accurately predicting latent states, even under challenging conditions.

**Weaknesses:**

- The model's assumptions regarding the linear relationship between demographics and reporting probabilities may be oversimplified for real-world applications.
- The application focuses on a specific case study (NYC 311), and it is unclear if the model performs equally well in different urban contexts or with different types of incident data.

**Questions:**

- Have you considered non-linear models for the relationship between demographics and reporting probabilities? Would this improve your predictions?
- How does your model perform in smaller cities or areas with less dense data compared to New York City?
- How does your model work well when it changes levels of sparsity in the ground truth data?

---

### Official Review · Reviewer_VQ9i · 2024-11-04

**Soundness:** 2
**Presentation:** 2
**Contribution:** 1
**Rating:** 3
**Confidence:** 2

**Summary:**

This paper introduces a GNN-based method for addressing urban applications, which uses both sparse unbiased ground truth data and biased proxy data for model training and prediction. A case study of urban incidents prediction, based on the New York City 311 complaints data, is conducted to evaluate the model. In this case, the government inspection ratings and the resident reports are considered as the unbiased data and biased proxy data, respectively. The model uses GNN to learn node embeddings and estimate these ratings and report probabilities.

**Strengths:**

1.	Combining biased and unbiased data effectively for urban applications is a valuable research direction.


2.	If the data collected in this work could be made open source, it would benefit related urban analysis research works.

**Weaknesses:**

1.	This work lacks novelty and technical contribution. It merely trains a standard GNN model for prediction, without making methodological improvements.


2.	This paper provides limited insights. Many of the conclusions drawn in this paper are obvious and have been widely studied. For instance, training using the ground truth data improves the accuracy, and integrating proxy data is potential to enhance the generalization in tasks without ground truth data, are well-known concepts in machine learning. Furthermore, the paper fails to demonstrate that the employed method has advantages over existing spatiotemporal modeling approach.


3.	The reliability of the experimental evaluation is questionable. First, the authors mention that they split the dataset into training and test sets, without allocating a validation set for hyper-parameters selection. Second, in the experiments on synthetic data, the ratings $r$ is specifically generated (eq.(11)) based on the form of the prediction model (eq.(2)). It makes the conclusions of such experimental analyses difficult to generalize to real-world applications, where the relationship between the ground truth data and the proxy data is unknown.


4.	The evaluation is only conducted on the NYC 311 complaints dataset, which is insufficient to support authors’ claim that the proposed approach is broadly applicable to prediction tasks.

**Questions:**

Please refer to weakness.

Pls.
1.	What does the "correlation" metric specifically refer to in the experiments?

---

### Official Review · Reviewer_QB8n · 2024-11-06

**Soundness:** 2
**Presentation:** 3
**Contribution:** 2
**Rating:** 3
**Confidence:** 4

**Summary:**

To tackle the biases in the observed data for urban event predictions, this paper proposed a GNN-based method to make use of both sparse unbiased ground truth rating data and biased crowdsourced report data. Through experiments with both semi-synthetic data and real-world data from NYC, the proposed method demonstrates enhanced accuracy in predicting true ratings and effectively reproduces demographic-related biases presented in crowdsourced reports.

**Strengths:**

(1) Originality: This paper is unique in that it jointly uses sparse unbiased data and dense biased data to train GNNs for urban event forecasting.
(2) Quality: The proposed model is evaluated on both semi-synthetic data and real-world data, demonstrating increased prediction accuracy and capability to capture the reporting biases related to demographic features.
(3) Clarity: The presentation is generally clear and easy to follow, with proper usage of mathematical expressions and visualizations.

**Weaknesses:**

(1) Table 2 provides the impression that the full model is not significantly better than taking the reports-only model for predicting reports & the ratings-only model for predicting ratings. In other words, there seems to be no ‘1+1>2’ effects here. It is unclear what are the benefits brought by fusing these two sources of data into one model for predicting reports and ratings.
(2) Figures 2-4 only compares the full model with the reports-only model on predicting ratings breaking down by event types. It is not clear how the full model compares with the rating-only model on these dimensions.
(3) Some results are presented with inadequate considerations on the statistical significance: For example, the results are obtained across multiple synthetic datasets, so I would expect the report on deviations besides mean/median. Similarly, in Figure 2 (a) and Figure 3, it is a bit strange why the ‘Mean’ results are presented without error bars.

**Questions:**

(1) Corresponding to the first point in ‘Weaknesses’, what are the benefits brought by fusing these two sources of data into one model for predicting reports and ratings?
(2) Comparing the full model with the reports-only model, do the performance gains come solely from being able to access part of the true rating data? Because this may provide a lot more information on the true data distribution than the reports data even in the absence of data leakage.
(3) Corresponding to the second point in ‘Weaknesses’, how does the full model compare with the rating-only model on Figures 2-4?
(4) Corresponding to the third point in ‘Weaknesses’, I would suggest refine the tables and figures to include deviances/error bars.

---

### Author Response · Authors · 2024-11-15

Thank you to all the reviewers for your time! The feedback is invaluable and we'll certainly take these suggestions into account as we revise!

---

### Note · Authors · 2024-11-15

I have read and agree with the venue's withdrawal policy on behalf of myself and my co-authors.